# Bile Acids in Intrahepatic Cholestasis of Pregnancy

**DOI:** 10.3390/diagnostics12112746

**Published:** 2022-11-09

**Authors:** Maciej Majsterek, Magdalena Wierzchowska-Opoka, Inga Makosz, Lena Kreczyńska, Żaneta Kimber-Trojnar, Bożena Leszczyńska-Gorzelak

**Affiliations:** Chair and Department of Obstetrics and Perinatology, Medical University of Lublin, 20-090 Lublin, Poland

**Keywords:** bile acids, intrahepatic cholestasis of pregnancy, pregnancy

## Abstract

Intrahepatic cholestasis of pregnancy (ICP) is the most common, reversible, and closely related to pregnancy condition characterized by elevated levels of bile acids (BAs) in blood serum and an increased risk of adverse perinatal outcomes. Due to the complex interactions between the mother and the fetus in metabolism and transplacental BAs transport, ICP is classified as a fetal-maternal disease. The disease is usually mild in pregnant women, but it can be fatal to the fetus, leading to numerous complications, including intrauterine death. The pathophysiology of the disease is based on inflammatory mechanisms caused by elevated BA levels. Although ICP cannot be completely prevented, its early diagnosis and prompt management significantly reduce the risk of fetal complications, the most serious of which is unexpected intrauterine death. It is worth emphasizing that all diagnostics and management of ICP during pregnancy are based on BA levels. Therefore, it is important to standardize the criteria for diagnosis, as well as recommendations for management depending on the level of BAs, which undoubtedly determines the impact on the fetus. The purpose of this review is to present the potential and importance of BAs in the detection and rules of medical procedure in ICP.

## 1. Introduction

Intrahepatic cholestasis of pregnancy (ICP) is the most common liver disease that develops during pregnancy [1,2,3,4]. It is a condition specific for the period of pregnancy, which resolves relatively quickly and spontaneously after its completion, but may recur in subsequent pregnancies (up to 90% of subsequent pregnancies), often with a more intense course [3,5].

ICP is diagnosed on the basis of typical clinical symptoms, laboratory abnormalities, and differential diagnosis excluding other causes of skin pruritus and liver dysfunction in the pregnant woman. The disease is not associated with abnormalities detected in imaging owing to the fact that biliary ducts are not dilated and hepatic parenchyma appears normal. Its dominant symptoms are skin pruritus and increased levels of bile acids (BAs) in the blood serum of the pregnant woman [6]. An increase in serum total BA concentration is the key laboratory finding, which allows for identification of the disease as it is present in over 90% of affected pregnancies. The physical examination does not reveal any primary skin lesions, but traces of scratching and prurigo nodules secondary to scratching may be visible. Jaundice occurs within one month of the onset of pruritus in 14–25% of patients [7]. Nevertheless, jaundice as the only symptom prompts the search for other causes.

Currently, researchers believe that the diagnosis of the disease should be made on the basis of abnormalities in liver function tests (LFTs), with or without itching, in the pregnant woman [8,9,10]. Although ICP cannot be completely prevented, its early diagnosis and prompt management significantly reduce the risk of fetal complications, the most serious of which is unexpected intrauterine death. The purpose of this review is to present the potential and importance of BAs in the detection and rules of medical procedure in ICP.

## 2. Pathophysiology, Etiology, and Complications of ICP

ICP is a disorder that is potentially harmful to the fetus. A clear relationship between elevated BA levels in maternal serum and fetal disorders has been confirmed in clinical practice, but the underlying mechanisms remain uncertain. One of these mechanisms is undoubtedly the inflammatory mechanism that underlies the pathophysiology of ICP [11,12,13,14]. Elevated levels of BAs induce the production of pro-inflammatory mediators in hepatocytes, attracting immune cells and initiating inflammation in the liver, eventually leading to cholestatic liver damage [15]. Furthermore, besides direct cytotoxic liver damage from BAs, oxidative stress and BA-induced mitochondrial damage lead to an inflammatory cascade [13,16]. The NLR family pyrin domain containing 3 (NLRP3) inflammasome in hepatic stellate cells and Kupffer cells is activated by BAs, causing inflammation or fibrosis [16]. The level of intracellular γ-glutamyl-L-cysteinyl-glycine (GSH), which protects cells against oxidative stress, cell proliferation, and division, is significantly influenced by inflammation and oxidative stress related to cholestasis [17]. Activation of the NF-κB pathway through the G protein-coupled BA receptor 1 (GPBAR1) is induced by BAs, resulting in elevated levels of inflammatory genes in trophoblasts, abnormal leukocyte infiltration, and placental inflammation [12]. ICP, by altering the metabolism of BAs and the fetal intestinal microflora, may increase the offspring’s susceptibility to inflammation [11]. Lin et al. [11] suggested that supplementation with *Lactobacillus rhamnosus* LRX01 may improve intestinal immunity in ICP offspring by inhibiting the expression of farnesoid X receptor (FXR) in the ileum.

ICP is a disease that is characteristic of the second half of pregnancy, especially the late second and entire third trimester of pregnancy. Despite this, cases of ICP already developing in the first trimester of pregnancy have been described in the literature [8,18,19,20,21,22]. The reported cases were associated with the hyperstimulation of the ovaries after in vitro fertilization.

The disease occurs at different rates; the average incidence of ICP remains at the level of 0.2–15.6%, but shows large ethnic, geographic, and seasonal variation [23]. The highest incidence of ICP is recorded in South American countries, especially Chile and Bolivia [24,25].

The etiology of ICP is complex and not fully explained. It is believed that the development of ICP is the result of many factors, such as: genetic predisposition, hormonal factors, environmental factors and nutritional deficiencies, and the influence of chronic diseases on the predisposition to developing cholestasis.

Seasonal incidence was noted in several works; it is more common in the winter months [8,26,27,28]. In addition, the contribution of many other factors is emphasized, such as: multiple pregnancies, assisted reproductive techniques, gestational age >35 years [29], and the coexistence of cholelithiasis or hepatitis C virus infection, affecting the more frequent development of the disease during pregnancy [30]. Noteworthy is the significantly higher incidence of the disease both in twin pregnancies (21%) [31] and in pregnancies obtained by in vitro fertilization compared to pregnancies obtained by natural conception (2.7% vs. 0.7%) [32]. Low selenium levels and low vitamin D levels may be potential causal factors, although specific environmental factors have not been identified [33,34].

The disease is usually mild in pregnant women, but it can be fatal to the fetus, leading to numerous complications, including intrauterine death [35]. The main complications related to this disease are presented in Figure 1. They include, but are not limited to, neonatal respiratory distress syndrome (associated with the presence of BA in the lungs), meconium-stained amniotic fluid, preterm birth, and an increased risk of stillbirth [36,37,38,39,40,41,42,43].

Fetal death in ICP may be caused by sudden vasoconstriction of the placental surface vessels or the development of arrhythmias induced by elevated BA levels, although pathophysiology has not been confirmed [38,39,40].

## 3. Bile Acids

The adult human body is characterized by the presence of numerous endogenous substances, including bile pigments (bilirubin, biliverdin), Bas, and countless xenobiotic substances. All these substances are transformed and eliminated mainly through the hepatobiliary system [44].

The total pool of BAs is the sum of the concentration of all BAs produced in the human body, including cholic acid (CA), chenodeoxycholic acid (CDCA), deoxycholic acid (DCA), ursodeoxycholic acid (UDCA), lithocholic acid (LCA), and hyodeoxycholic acid (HDCA). Primary BAs, CA, and CDCA are the most concentrated in the blood serum, while the remaining secondary and tertiary BAs are found in very small amounts in both healthy subjects and women with ICP.

Disturbances in BA homeostasis may arise from genetic mutations responsible for the transport and synthesis of these molecules, impairing hepatocyte function or intestinal changes that may be associated with intrahepatic cholestasis, gallbladder disease, glucose intolerance, and non-alcoholic and alcoholic fatty liver disease [45].

Fasting concentration of BAs is a sensitive marker of liver disease [46,47]. Postprandial assessment of BAs level was characterized by greater diagnostic sensitivity, while the determination of BAs in the fasting state was characterized by greater specificity [48]. Currently, due to greater diagnostic specificity, it is proposed to determine the level of BAs, both in pregnant and other patients, on an empty stomach [49,50].

### 3.1. Synthesis and Enterohepatic Circulation of Bile Acids

BAs are synthesized in the liver and constitute the major end product of cholesterol catabolism which proceeds by a multi-enzymatic pathway involving at least 17 different enzymes. In an adult human, approximately 500 mg of cholesterol is converted into BAs per day [51,52]. The rate of BA biosynthesis in the initial stage is partially limited by cholesterol 7α-hydroxylase, an enzyme from the cytochrome P450 family. The expression of the CYP7A1 gene encoding cholesterol 7α-hydroxylase, as well as the activity of the enzyme itself, are subject to strict, multifactorial regulation, including such factors as: BAs concentration, hormones (including insulin, glucagon, glucocorticosteroids), cholesterol (oxysterols), cytokines, and the daily cycle [45,53,54,55]. During the synthesis process, primary BAs, CA, and CDCA are first formed, which in the final stage are coupled with the amino acids: glycine or taurine (in a ratio of approximately 3:1). As a result of the conjugation process, primary BAs lose their ability to cross cell membranes. The process of bile formation is a process that requires the participation of energy from the breakdown of ATP (adenosine-5′-triphosphate) and consists of the transport of bile components through cell membranes, from hepatocytes to the bile ducts, against their concentration gradient. BAs are cleared into the bile ducts by a specific ATP-dependent transporter, bile salt export pump (BSEP), a product of the ABCB11 gene, which is highly specific and only transports conjugated BAs. BAs are the main component of bile, and their concentration in bile can be up to 1000 times higher than in the interior of the hepatocyte [56]. The second major component of bile is phosphatidylcholine (PC), which is excreted into the biliary tract by the ATP-dependent multidrug resistance protein (MRP) 3 transporter, a product of the ABCB4 gene. Phosphatidylcholine plays an important, protective role in the interstitial space. During bile formation, BAs transported by BSEP form mixed micelles with phosphatidylcholine. These complexes have a protective effect on the epithelium lining the bile ducts against the toxic and detergent effects of bile salts and thus allow their secretion without damaging the surrounding cells. Phosphatidylcholine secretion, in parallel with bile salts, is necessary to maintain adequate bile flow [57].

In addition to the two main components, bile also contains numerous conjugated organic ions that are excreted from hepatocytes via the MRP2 transporter, a product of the ABCC2 gene [45,58]. These substances include, but are not limited to, bilirubin, conjugated drug metabolites, and many other substances. The heterodimeric protein complex, consisting of the two membrane proteins, namely, ABCG5 and ABCG8, is also involved in cholesterol transport. Furthermore, MRP1, which is a product of the ABCB1 gene, which is a homologue of MRP2mrp, but with much broader substrate specificity, is involved in the transmembrane transport of conjugated drug metabolites.

Bile produced in the liver is accumulated in the gallbladder until it is released into the gastrointestinal tract under the influence of postprandial cholecystokinin release—a peptide hormone that initiates postprandial contraction of the gallbladder [59]. In the small intestine, BAs emulsify dietary fats, fat-soluble vitamins, and other lipids. Under the influence of the anaerobic bacterial flora in the intestine, a number of primary BA transformations occur, including deconjugation and dehydroxylation, which lead to the formation of secondary BAs, i.e., DCA and LCA [60,61,62,63,64]. As a result of further changes, under the influence of both hepatic and intestinal mechanisms, substances of minor importance are formed, i.e., tertiary BAs: HDCA and UDCA. Through specific protein transport systems in enterocytes, bile salts are reabsorbed and reach the liver via the portal vein and are taken up by hepatocytes within the sinusoidal membrane [58,65].

Enterohepatic circulation is extremely efficient; the liver takes up about 95% of BAs and the remaining 5% is excreted in the feces. This loss is replaced by de novo synthesis of BAs in the liver [51,66].

### 3.2. The Biological Role of Bile Acids

The basic function of BAs in the human body is above all participation in digestive processes, and additionally participation in other physiological processes occurring in the human body [66]. These processes include the emulsification and absorption of lipids and lipophilic vitamins contained in food, and the absorption of calcium. In combination with phospholipids, they form complexes that facilitate the dissolution of cholesterol and other lipids in bile. BAs affect the secretion of pancreatic enzymes and cholecystokinins. The osmotic pressure gradient that arises during the secretion of BAs into the bile ducts is one of the most important factors ensuring the proper flow of bile through the liver [45,57]. Due to their detergent properties, BAs can damage the cell membranes of the biliary epithelium and, consequently, damage the liver parenchyma [67,68,69,70].

Originally, four major functions of BAs were identified:
constitute the main important mechanism for the elimination of excess cholesterol through their synthesis and subsequent fecal excretion.BAs and phospholipids prevent cholesterol from precipitating in the gallbladder by dissolving cholesterol in the bile.they act as emulsifiers, increasing the availability of fats for pancreatic lipases, facilitating the digestion of triacylglycerols in the diet.they enable the intestinal absorption of fat-soluble vitamins [56,67,69].

Nevertheless, in recent years, new insights into the biological activity of BAs have been confirmed. The latest discoveries have confirmed that BAs are involved in regulating their own metabolisms, transport through enterohepatic circulation, in glucose, lipid and overall energy expenditure metabolism, and in controlling signaling events in liver regeneration [71,72].

After the characterization and isolation of FXR, for which bile acids are physiological ligands, the functions of Bas in the regulation of glucose and lipid homeostasis were confirmed. As indicated above, BAs binding to FXR impair the expression of genes that participate in overall BA homeostasis (e.g., FGF19). However, genes that participate in BA metabolism are not the only ones that are controlled by FXR action as an effect of binding BA [73,74,75].

FXR controls genes that metabolize in the liver lipids (e.g., SREBP-1c), lipoproteins (e.g., apoC-II), glucose (e.g., PEPCK), and that are involved in hepatoprotection (e.g., CYP3A4, which is nifedipine oxidase).

Furthermore, except for their roles in activating FXR and lipid emulsification in the intestine, the BAs assist in various signal transduction processes through the mitogen-activated protein kinase (MAPK) pathways as well as activation of the c-JUN N-terminal kinase (JNK). The pregnane X receptor (PXR), the vitamin D receptor (VDR), and the constitutive androstane receptor (CAR) are other members of the BAs-activated nuclear receptor family. An additional receptor activated in response to BAs is the G-protein coupled BA receptor 1, GPBAR1, which might be involved in the control of obesity. GPBAR1 activation in brown adipose tissue triggers the uncoupling of protein 1, thermogenin (UCP1), which leads to increased energy expenditure [73,74,75,76].

### 3.3. Bile Acids in the Fetus

During intrauterine life, due to the anatomical and functional immaturity of the developing fetus’ liver, there are some functional and structural differences in the transformation and elimination of harmful substances. The main organ responsible for metabolism and elimination of metabolic products during fetal intrauterine life is the placenta [77,78,79,80]. The processes in the placenta are significantly similar to those in the liver of an adult human.

In utero, hepatic biosynthesis of BAs and bilirubin begins relatively early, and BAs themselves reach relatively high concentrations [81,82]. The dominant BA present in fetal serum is CDCA, while CA is present in lower concentrations. It has been observed that already in the 12th week of pregnancy, the CA/CDCA bile acid concentration ratio is 0.85, around the date of delivery (38–40 weeks of pregnancy) it is 1.9, in the neonatal period it is 2.5, and in an adult 1.6 [83]. In addition, more BAs unbound or bound to glycine than to taurine are observed in fetal serum compared to adult serum. The above differences in the composition and concentration of BAs in the fetus, compared to adults, are caused by the immaturity of the enzyme systems involved in the metabolism of BAs in the fetus and their selective, transplacental exchange.

The process of BA synthesis in the fetus develops before the fetus has developed mechanisms that enable the effective secretion of BAs into bile. As a result, most of the synthesized BAs pass into the fetal blood serum, then a small fraction is excreted via the fetal kidneys into the amniotic fluid, and the remainder, the greater part, is eliminated through the placenta into the mother’s body. Most of the fetal pool of BAs is eliminated from the mother’s body via the gastrointestinal tract [84,85]. The identified factors influencing the maintenance of BA balance in the maternal-fetal circulation include: activity of hepatic metabolic pathways involved in the biosynthesis and transformation of BAs in both the mother and the fetus, the rate of BA elimination from the maternal organism, and the transport properties of the placenta [81,86].

### 3.4. Bile Acids in Physiological Pregnancy

In physiological pregnancy, the transplacental flow of BAs is supported by a gradient of BAs and bicarbonate concentrations and proceeds from the fetus to the mother [6,87]. This process depends on the protein transporters from the organic anion transporting polypeptides (OATP) family and ATP-dependent protein carriers from the ATP—binding cassette (ABC) family [58,88]. Simple diffusion, despite the possible free two-way flow of BAs through the placenta, does not appear to be the main mechanism responsible for the transport of BAs.

In physiologically running pregnancy, a slight increase in the total concentration of BAs in the blood serum is observed with the advancement of pregnancy. In studies that assessed the level of individual BAs, it was found that the concentration of secondary BAs did not change significantly, while the level of CDCA doubled around the time of delivery [89]. The data on CA levels are inconclusive as some studies have shown a significant increase in CA levels in the third trimester compared to the first trimester, while others have not shown changes in CA levels [89,90]. Despite this, the level of CA in all studies remained within the normal range.

The literature also describes the phenomenon of asymptomatic hypercholanemia of pregnancy (AHP), which is defined as an increased concentration of BAs in the serum of a pregnant woman without clinical symptoms and no other laboratory abnormalities [91]. AHP may affect up to 10% of pregnant women, and in this group approximately 2–3% of pregnant women may develop ICP in the second trimester of pregnancy. In AHP, the most significant change in BAs’ profile is increased CA concentration with relatively unchanged CDCA level [92].

## 4. Bile Acids in Pregnancy Complicated by ICP

The most important role in the pathomechanism of ICP is played by the increased concentration of BAs and their detergent properties, which accumulate in hepatocytes, causing damage to cell membranes and the release of aminotransferases, bilirubin, γ-glutamyl transpeptidases (GGTP), and alkaline phosphatase into the blood serum.

Nevertheless, it has been shown that elevated, >10 µmol/L, postprandial BA values are observed in up to 40% of asymptomatic pregnancies, which supports the theory that pregnancy is a cholestatic state (hypercholanemia) [49]. The achieved BA values in ICP can reach up to 100 times the reference values. Thus far, there is no consensus as to whether an increase in serum BAs precedes clinical symptoms. There have been reports of BAs rising prior to the onset of clinical symptoms or the appearance of other biochemical abnormalities [86,93].

Most guidelines agree that typical clinical symptoms include pruritus of the skin, which is frequently generalized, but commonly begins and predominates on the palms and soles. The itching sensation strengthens at night, oftentimes involves the right upper quadrant pain and may be accompanied by nausea, poor appetite, sleep deprivation, or steatorrhea. Abnormalities in the biochemical functions of the liver should be identified in the absence of diseases with similar clinical symptoms and laboratory abnormalities [94]. However, the type and reference values of laboratory markers of liver dysfunction that should be considered diagnostic for the diagnosis of ICP may differ significantly between societies [94,95,96,97,98,99,100,101,102].

Pregnant skin pruritus is accompanied by elevated levels of hepatic function tests or BAs, not caused by other diseases, which normalize after delivery. In addition, the Royal College of Obstetricians and Gynaecologists (RCOG) guidelines emphasize that in the presence of typical clinical symptoms and abnormal levels of hepatic function tests, an increased concentration of BAs is not necessary for the diagnosis of ICP, and the definitive confirmation of the diagnosis is the resolution of clinical symptoms and normalization of laboratory markers of liver function after delivery [98]. It is important to repeat the laboratory tests every week when the initial level of total aminotransferase and BAs are normal due to the fact that pruritus may precede an increase in serum BAs by several weeks. However, if UDCA becomes empirical, aminotransferases and BAs may never increase. Nevertheless 23% of pregnancies are affected by pruritus, but only a minority are caused by ICP.

In a pregnancy with ICP, the BAs flow is reversed from mother to fetus, leading to an increase in the fetal pool of BAs. The above observation may be confirmed by the results of the study conducted by Laatikainen, in which the author found a higher concentration of BAs in the umbilical cord blood of fetuses from ICP pregnancies compared to those from uncomplicated pregnancies (5.6 vs. 1.8 µg/mL) [103]. Macias et al. showed that the cause of the BAs flow reversal is structural and functional changes developing within the placenta in the course of ICP [104]. Another study found that impaired transplacental BA elimination was also the cause of the increase in the concentration of BAs in umbilical cord blood [105]. In an advanced pregnancy, when the mother’s liver is exposed to high levels of sex hormones, a physiological increase in the level of BAs in both the mother and the fetus, with abnormal placental function leading to a loss of the ability to synthesize water-soluble BAs, may lead to disturbances in transplacental BA transport and ICP development [106].

The accumulation of BAs in the maternal and fetal liver observed in ICP may lead to oxidative stress and hepatocyte apoptosis, significantly affecting the course of pregnancy [6,107,108].

An altered BA profile is observed in women with ICP. CA remains the major BA and its concentration is significantly higher than that of CDCA, resulting in an increase in the CA/CDCA ratio and a decrease in the percentage of CDCA in the total pool of BAs. There is also an increase in the level of secondary BAs, mainly DCA, less than that of the primary BAs, which may indicate an impairment of their enterohepatic circulation [109]. In addition, ICP shows an increase in taurine-conjugated BAs and a decrease in glycine-conjugated BAs, resulting in a lowered glycine/taurine ratio. Changes in BAs in pregnancy complicated by ICP are shown in Figure 2.

The level of total serum BAs is currently considered to be the most sensitive and specific biochemical marker used in the diagnosis and monitoring of ICP [94]. Some researchers, in order to increase the specificity of ICP recognition, were in favor of extending the above diagnostics to the assessment of the concentrations of individual BAs, including the level of CA or the CA/CDCA concentration ratio, especially at an early stage of diagnosis [27,90]. However, it has been shown that the assessment of the concentration of total BAs in the blood serum is characterized by a similar sensitivity and specificity in the diagnosis of ICP compared to the assessment of the CA/CDCA ratio [43].

There are no uniform ICP diagnostic criteria among the guidelines of various scientific societies [94,95,96,97,98,99,100,101,102]. The reference range of serum BAs’ concentration used in the diagnosis of ICP is different (Table 1).

## 5. Management of ICP

The main purpose of the therapy of ICP is to reduce the severity of clinical symptoms and normalize the biochemical exponents of maternal liver dysfunction and minimize the risk of complications, both fetal and neonatal.

The treatment of ICP is based on various therapeutic options, which include non-pharmacological treatment, the selection of appropriate pharmacotherapy, and the choice of the method and time of delivery.

Non-pharmacological treatment emphasizes the important role of bed regimen, which leads to improvement of hepatic blood flow, monitoring the well-being of the fetus by using biophysical methods (cardiotocography, ultrasound examination), assessment of fetal movements by the pregnant woman, and monitoring of concentrations of hepatic functional tests. Many scientific societies recommend monitoring total bile acid concentrations, with specific recommendations varying from one society to another. The RCOG recommends determination of bile acid concentrations at weekly intervals [98]. The Team of Experts of the Polish Gynecological Society (PGS) presented a similar position to the RCOG and recommends determining bile acid levels at least once a week [101]. Whereas the Society for Maternal-Fetal Medicine (SMFM) does not present detailed recommendations on the frequency of BA determination [99].

The decision on the timing and method of delivery in pregnancies complicated by intrahepatic cholestasis of pregnancy should take into account the advancement of pregnancy, BA levels, clinical signs and obstetric history, and should be made promptly in case of either fetal, maternal, or both, emergency:
at the concentration of <40 µmol/L total bile acids, childbirth is recommended between 37 0/7 and 38 6/7 weeks of pregnancy.at the concentration of 40–99 µmol/L total bile acids, childbirth is recommended between 36 0/7 and 37 0/7 weeks of pregnancy.at the concentration of >100 µmol/L total bile acids, childbirth is recommended in the 36 0/7 week of pregnancy, or earlier in the case of other events or emergency to either the fetus, mother, or both [110].

UDCA is the main medication in the pharmacotherapy of intrahepatic cholestasis in pregnancy. It is a naturally occurring, hydrophilic, tertiary bile acid formed in the gastrointestinal tract through the bacterial metabolism of primary bile acids. After oral use the maximum blood concentration is reached after about 30–60 min. The mechanism of action of UDCA is multidirectional:contributes to the break-up of micelles in the intestines and reduces the rate of cholesterol absorptionreduces hepatic synthesis and secretion of cholesterol,has a choleretic effect,has an anti-apoptotic effect,corrects the disturbed kinetics of transplacental bile acid transport in pregnancies complicated with intrahepatic cholestasis during pregnancy,reduces the concentration of primary bile acids in the umbilical cord blood and in the amniotic fluid.dosing is 16–21 mg/kg body weight, usually spread over 3 doses [47,105,111].

Medicaments used in the treatment of intrahepatic cholestasis in pregnancy also include: ornithine aspartate, phospholipids, dexamethasone, cholestyramine, S-adenosine-L-methionine, antihistamines, and vitamin K.

## 6. Conclusions

ICP is the most common liver disease in pregnancy. ICP is a condition in which the dominant symptoms are skin pruritus and increased levels of BAs in the blood serum of the pregnant woman. The disease is mild in pregnant women, but it can be fatal to the fetus, leading to numerous complications, including intrauterine death. According to many clinical studies the mechanisms by which BAs affect the fetus during pregnancy are fairly well understood, but still require more in-depth attention. It is worth emphasizing that BAs are of limited importance in medical diagnostics, with the exception of ICP, on which all diagnostics as well as pregnancy management are based. Therefore, it is important to standardize the criteria for diagnosis, as well as recommendations for management depending on the level of BAs, which undoubtedly determine the impact on the fetus. Hence, scientists agree that more research is needed to translate it into therapeutic strategies. Multicenter and controlled trials with more patients are required to compare bile acid levels with fetal complications as there is increasing evidence of BAs being used as a prognostic test in women with ICP.

## Figures and Tables

**Figure 1 diagnostics-12-02746-f001:**
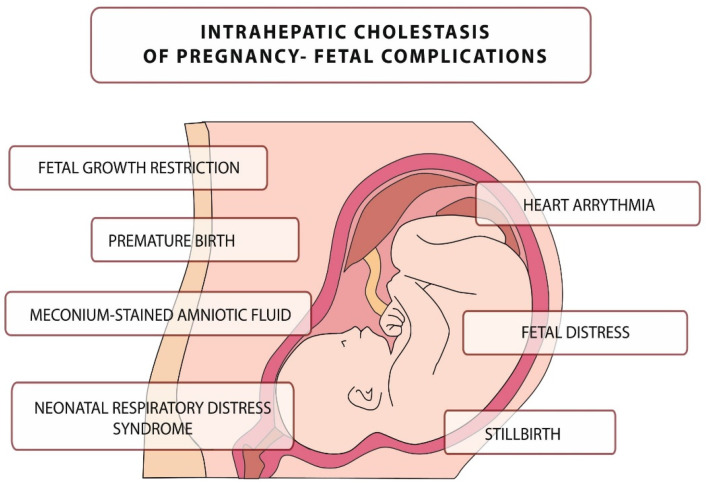
Intrahepatic cholestasis of pregnancy–fetal complications.

**Figure 2 diagnostics-12-02746-f002:**
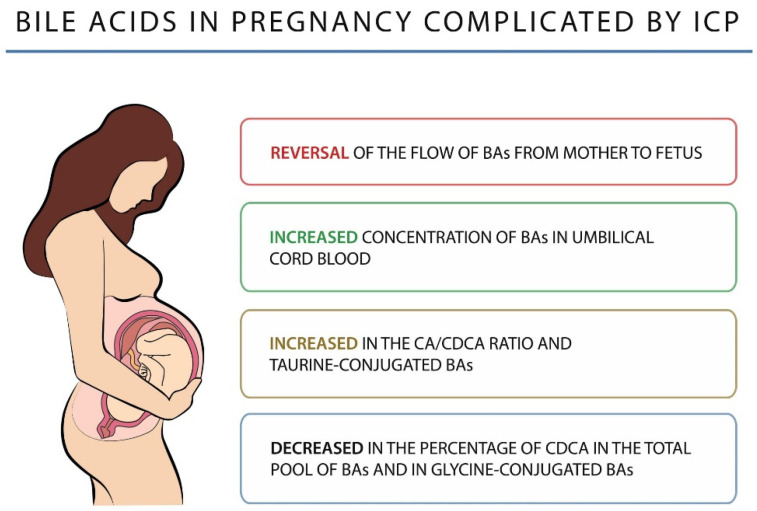
Bile acids in pregnancy complicated by ICP.

**Table 1 diagnostics-12-02746-t001:** The reference range of serum BAs’ concentration in the diagnosis of ICP.

Guidelines	Bile Acids	Reference
RCOG	>19 μmol/L	[98]
SMFM	>10 μmol/L	[99]
SAPPG	>19 μmol/L	[100]
PGS	>10 μmol/L	[101]
ChSH and ChMA	>10 μmol/L	[102]

Royal College of Obstetricians and Gynaecologists (RCOG); Society for Maternal-Fetal Medicine (SMFM); South Australian Perinatal Practice Guideline (SAPPG); Polish Gynecological Society (PGS); Chinese Society of Hepatology (ChSH); Chinese Medical Association (ChMA).

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
