# Peer review of "Bile Acids in Intrahepatic Cholestasis of Pregnancy"

_diagnostics, 2022, doi:10.3390/diagnostics12112746_

Round 1
Reviewer 1 Report
The present manucript is well-written and comprehensive, th authors have written a narrative review on the IHCP. I have some minor recommendations for the authors before the manuscript is considered for publication in this journal. IHCP was found to be closely associated with inflammation. In my opinion the pathophysiology of altered immune response and its impact on the developing fetus should be discussed in more detail. Possible effect of inflammation together with the bile acids should be stated. Treatment options and the monitoring for treatment success should be underlined.
Author Response
Dear Reviewer 1,
We would like to resubmit our manuscript entitled “Bile acids in intrahepatic cholestasis of pregnancy”. We appreciate your valuable remarks and hope that the quality of our manuscript is going to meet your expectations now that we have made some suggested alternations.
The present manuscript is well-written and comprehensive, th authors have written a narrative review on the IHCP. I have some minor recommendations for the authors before the manuscript is considered for publication in this journal.
Thank you very much for finding the time to read our manuscript. Thank you for considering our manuscript. We have corrected the manuscript point by point according to your suggestions.
IHCP was found to be closely associated with inflammation. In my opinion the pathophysiology of altered immune response and its impact on the developing fetus should be discussed in more detail. Possible effect of inflammation together with the bile acids should be stated.
Thank you for such helpful remark. We have added sentences related to this aspect.
Treatment options and the monitoring for treatment success should be underlined.
Following your advice, we have introduced the part entitled “5. Management of ICP” and additional references.
Yours faithfully,
Zaneta Kimber-Trojnar
Chair and Department of Obstetrics and Perinatology
Medical University of Lublin, 20-090 Lublin, Poland
Tel: +48-81-7244-769
E-mail: [email protected]

Reviewer 2 Report
The title is clear and concise. The abstract represents the manuscript content, however, looks too narrow and must be improved to represent a review article that is supposed to provide an expert knowledge content.
The manuscript should be formatted according to the journal requirements. Authors affiliations are required.
Turnitin's check shows a similarity index of 55%, which is extremely high.
The overall manuscript structure should be reconsidered. The introduction part is too long. In the introduction, part I would suggest providing definition(s), epidemiology and justify importance of the topic discussion. Classification, etiology, symptoms, and risk factors of the condition(s) you are discussion in the review should be presented in the separate section.
Numbering of sections and subheadings are wrong. Please check it carefully.
A very important section on management of ICD is missing, however, is mentioned in the abstract. Please create a section to discuss currently approved management approaches and guidelines.
Author Response
Dear Reviewer 2,
We would like to resubmit our manuscript entitled “Bile acids in intrahepatic cholestasis of pregnancy”. Thank you very much for finding the time to read our manuscript. Thank you for considering our manuscript. We have corrected the manuscript point by point according to your suggestions.
The title is clear and concise. The abstract represents the manuscript content, however, looks too narrow and must be improved to represent a review article that is supposed to provide an expert knowledge content.
Following your advice, we have re-edited the article abstract.
The manuscript should be formatted according to the journal requirements. Authors affiliations are required.
Thank you for your suggestion. The corresponding modifications have been made.
Turnitin's check shows a similarity index of 55%, which is extremely high.
Thank you for your comment. We apologize for the confusion. The sentences have been corrected.
The overall manuscript structure should be reconsidered. The introduction part is too long. In the introduction, part I would suggest providing definition(s), epidemiology and justify importance of the topic discussion. Classification, etiology, symptoms, and risk factors of the condition(s) you are discussion in the review should be presented in the separate section.
Thank you for your suggestion. Appropriate modifications were made.
Numbering of sections and subheadings are wrong. Please check it carefully.
Thank you for such helpful remark. We have corrected the numbering of sections and subheadings in accordance with the “Instructions for authors”.
A very important section on management of ICD is missing, however, is mentioned in the abstract. Please create a section to discuss currently approved management approaches and guidelines.
Following your advice, we have introduced the part entitled “5. Management of ICP” and additional references.
Yours faithfully,
Zaneta Kimber-Trojnar
Chair and Department of Obstetrics and Perinatology
Medical University of Lublin, 20-090 Lublin, Poland
Tel: +48-81-7244-769
E-mail: [email protected]

Round 2
Reviewer 2 Report
Thank you very much for addressing the comments.